# The Relevance of Noncoding DNA Variations of Paraoxonase Gene Cluster in Atherosclerosis-Related Diseases

**DOI:** 10.3390/ijms22042137

**Published:** 2021-02-21

**Authors:** Anna Wysocka, Agnieszka Zwolak

**Affiliations:** 1Chair of Internal Medicine and Department of Internal Medicine in Nursing, Faculty of Health Sciences, Medical University of Lublin, 20-093 Lublin, Poland; katedrainterny@umlub.pl; 2Chair and Department of Cardiology, Medical University of Lublin, 20-954 Lublin, Poland; 3Chair and Department of Endocrinology, Medical University of Lublin, 20-954 Lublin, Poland

**Keywords:** paraoxonase, polymorphism, atherosclerosis

## Abstract

The human paraoxonase (*PON*) gene cluster is comprised of three contiguous genes (*PON1, PON2* and *PON3*) of presumably common origin coding three lactonases of highly similar structure and substrate specificity. The catalytic activity of PON proteins is directed toward artificial organophosphates and in physiological conditions toward thiolactones and oxidized phospholipids. Consequently, PON enzymes are regarded as an effective defense against oxidative stress and, as a result, against atherosclerosis development. Additionally, both PON’s serum activity and its concentration are influenced by several polymorphic variations in coding and noncoding DNA regions of the *PON* gene cluster remaining in linkage disequilibrium. Hence, the genetic polymorphism of the *PON* gene cluster may contribute to atherosclerotic process progression or deceleration. In this review the authors analyzed the relevance of noncoding DNA polymorphic variations of *PON* genes in atherosclerosis-related diseases involving coronary and peripheral artery disease, stroke, diabetes mellitus, dementia and renal disease and concluded that the effect of PON gene cluster’ polymorphism has a considerable impact on the course and outcome in these conditions. The following *PON* genetic variations may serve as additional predictors of the risk of atherosclerosis in selected populations and individuals.

## 1. Introduction

The paraoxonase gene cluster consists of three genes (*PON1, PON2, PON3*) localized on the long arm of chromosome 7 between q21.3 and q 22.1 [1] and comprises summarily 136 KB arranged in order *PON1, PON3* and *PON2* [2]. All three genes exhibit a high homology of structure and probably evolved from a common precursor. The *PON2* gene is presumably the oldest gene in the *PON* cluster; after which, via duplication, *PON3* appeared, and finally *PON1* arose [3]. Based on the structural homology and similar range of substrates, the lactonohydrolase AKU 3702 of the fungus *Fusarium oxysporum* is considered to be their common evolutionary ancestor [4]. Interestingly, enzymes acting like PONs were found in different species of animals or plants and additionally an appreciable identity of *PON* genes between various mammalian species was detected [5,6]. These findings may indicate how important the physiological role of PONsis.

The *PON1* gene is expressed in the liver of adults as well as in the spleen and the liver of fetuses, while the PON1 protein is found mainly in plasma tightly bound to high density lipoprotein (HDL) [7], similarly to PON3 [8]. The *PON2* gene is expressed in many different tissues and the protein is not found in plasma but in the mitochondria and the endoplasmic reticulum of the liver, brain, kidneys, heart, pancreas and intestine as well as the endothelial and smooth muscle cells of the aorta [9].

The common name of the family of enzymes coded by *PON* cluster gene— paraoxonases (PON1, PON2, PON3)—derives from the first studies regarding PON1, when the activity of the enzyme towards toxica oxon metabolite (paraoxon) was detected [10]. According to Aldridge, paraoxonases were included into class A esterases hydrolyzing organophosphorous compounds, which is the opposite to B class esterases, which are irreversibly inhibited by organophosphates (OP) [11]. In the modified Enzyme Classification of Enzymatic Committee of International Union of Biochemistry and Molecular Biology, paraoxonases have been classified at number E.C 3.1.8.1 as aryldialkyl phosphatases [12]. Successively, the broad substrate specificity of PON1, including organic phosphates, aryl esters of carboxylic acids, aromatic carbons, lactones and unsaturated fatty acids esters, was described [13,14]. In spite of lacking OP-hydrolyzing activity, the later discovered PON2 and PON3 were termed by analogy with paraoxonases. All three enzymes are characterized by lactonase activity including the ability to hydrolyze homocysteine (Hcy) thiolactone [15] and acyl-homoserine lactones of bacterial quorum sensing factor [16].

Numerous studies suggest that due to their enzymatic activity of hydrolyzing peroxidized phospholipids and Hcy thiolactone, PONs play an important antioxidant function and are responsible for the protective properties of HDLs against atherosclerotic plaque formation; however, the exact mechanisms of these phenomena are not completely explained [17,18]. Additionally, it was observed that decreased activity of PON1 enhances the progress of atherosclerosis [19] and may indicate the risk of atherosclerosis-related disease occurrence [20] and severity [21]. Moreover, increased PON1 activity improves prognosis in patients with severe overt symptoms of atherosclerosis [22].

It is known that PON1 activity is highly influenced by genetic polymorphism, but it may also be modulated by numerous pharmacological, nutritional and lifestyle factors [23]. It results in as much as a40-fold variation in enzyme activity between individuals, as well as considerable differences between different populations [24,25,26]. The most intensively examined polymorphisms apply to the *PON1* coding region Q192R and L55M, which significantly affect PON1 activity. Additionally, several SNPs was revealed in both the 5′ and 3′ untranslated regions (UTRs) of the *PON1* gene as well as in its introns. Amongst the most widely investigated promoter polymorphic sites, including -107/108(C/T), -126(G/C), -160/-162(A/G), -824/832(A/G) and -909 (G/C), only -108(C/T) and -162(A/G) are known to be functional [27,28,29], with the greatest effect being caused by the -108(C/T) polymorphism in the promoter region. Two polymorphic sites described in the coding region of the *PON2* gene are situated in codon 148(A/G) and 311(C/S) [9] and two functional SNPs of noncoding DNA regions involve the A/C variant located in intron-1 (rs12794795) [30] and T/A variant at position rs2375005 [31]. In the *PON3* gene several SNPs were detected, localized in the coding region at positions 107(G/A) and 99(G/A), as well as 51 GG, 73 GG, 99 GG and 311 S/T, 324 G/D, being, respectively, silent and missense polymorphic variations [2,32,33]. Among the SNPs discovered in the promoter region of the *PON3* gene, -567(C/T), -665 A/G and -(746 C/T) significantly affect the enzyme levels; the functional role of -133(C/A) polymorphism is unclear [34], while -4105(G/A), -4970 (T/G), -4984(A/G) are not functional [35].

In the last few years some comprehensive reviews elaborating the role of each member of the PON family have been published [32,36,37,38,39], but the relationship between noncoding DNA polymorphic variations of *PON* genes and atherosclerosis was not separately considered. Taking into consideration that polymorphic variations of the *PON* cluster gene directly affect PONs’ activity, concentration and the binding regulatory elements it is reasonable to believe that they may contribute to atherosclerosis–related diseases, such as coronary artery disease, stroke, peripheral artery disease, diabetes mellitus, dementia and renal disease. Indeed, this issue is worth of discussing since both, the *PON* genotype and enzyme activity, may serve as possible predictors of the clinical forms of atherosclerosis [40].

## 2. Search Strategy and Data Extraction

Aiming to identify relevant articles, Pubmed database searching was performed. The terms “paraoxonase promoter” AND “atherosclerosis” were used, allowing us to obtain only 34 articles published to 2020. Thus, the search was broadened to include “paraoxonase polymorphism AND atherosclerosis”—as result 278 papers were obtained. Preliminarily, all abstracts from the second group were analyzed for their appropriateness for inclusion in this review and then the chosen full texts were studied. Aiming to focus on the clinical relevance of genetic variations in particular diseases, the terms “paraoxonase promoter” AND, respectively, “coronary heart disease”, “stroke”, “diabetes”, “PAD”, “dementia” and “renal disease” were also used. The language of the selected full-text articles was English. Additionally, relevant articles were selected from the reference lists of papers considered to be appropriate and from articles, in which the latestpapers were cited. The reviewed data have been arranged into three sections involving (1) the functional relevance of *PON* variations (2) the impact of *PON* variations on lipid profile (3) the clinical relevance of *PON* variations consisting of five subsections: (i) coronary heart disease, (ii) stroke, (iii) diabetes mellitus, (iv) peripheral artery disease, dementia and renal disease (congregated in one subsection because of the scarce number of appropriate publications) and (v) drug response (divided into two parts: (a) antiplatelet drugs and (b) statins). The review finishes with three conclusions summarizing the considerations.

## 3. Effects of Genetic Polymorphism on PONs Activity and Concentration

Despite the existing wide spectrum of PONs substrates, in laboratory practice the most commonly measured were the enzymes’ activities towards paraoxon (paraoxonase activity) and towards phenyl acetate (arylesterase activity), and less commonly towards diazoxon (diazoxonase activity) and towards lactones (lactonase activity).

PON enzyme activity is affected by both environmental and genetic factors. In view of the topic of the present review, only the last will be discussed, with the particular consideration of the role of the noncoding DNA variations. The coding region variations are mentioned because they were often found to be in the strong linkage disequilibrium (LD) with polymorphisms of the noncoding regions.

There are a lot of SNPs in the *PON1* gene, consisting of 26 KB and including nine exons, eight introns and the promoter region [41,42]. It results in the synthesis of altered forms of the enzyme and influences its activity and concentration. In the promoter region several regulatory sequences binding transcription factors, such as specificity protein 1 (Sp1), sterol regulatory binding protein 2 (SREBP2) or nuclear factors 1 and Y (NF-1 and NF-Y), were described [43,44,45], but no canonical TATA box was identified [41], In vitro it has been proved that the overexpression of Sp1 significantly enhances *PON1* promoter activity [46]. The transcription factor activated by the arylhydrocarbon receptor (AhR) also binds to the xenobiotic regulatory element (XRE) in the *PON1* promoter region [47].

The genetic variations in the coding region of the PON1 gene are only partially responsible for a significant variability in PON1 activity between individuals and populations. In 192RR homozygotes the hydrolyzing activity of PON1 towards OP compounds such as paraoxon, fenitrokson or chlorpyriphos is higher in comparison with QQ homozygotes [14,48]. The ability of PON1 isoforms to protect the lipoproteins against the peroxidation process is higher in QQ genotype carriers [49]. Alloenzymes QQ are also characterized by their greater effectiveness in hydrolyzing diazoxon and neurotoxic gases [13]. On the other hand, there is a group of substrates, including phenyl acetate, towards which the hydrolysis rate is independent of the genotype Q192R [50]. Initially, it was considered that 55 L/M polymorphism did not affect the PON1 activity [51]. However, it was later proved that MM homozygotes are less efficient in the hydrolysis of paraoxon than LL homozygotes [52] and that the higher activity of the enzyme in the occurrence of the 55L allele is related to the significantly higher level of both the PON1 mRNA and the protein [53]. Additionally, the existence of the contributable LD between 192R and 55L was reported [54].

Amongst *PON1* variations localized in the upstream regulatory region, the -107 C/T polymorphism is believed to play the main role in the individual variability of PON1 status. In the cell–culture experiment Brophy et al. evaluated promoter-region polymorphisms and identified increased PON1 expression if the -108 C, -162 A or -909 G allele occurred [29]. In the next study, involving a sample of the white population, these same authors discovered the significant effect of the *PON1* -108 site accounting for 22.8% of the variation in arylesterase activity, the lesser effect of the -162 site and no independent effect of the -909 site, being in linkage disequilibrium with the other sites. Besides the significant LD along the whole regulatory region, the authors also confirmed the LD of the 55L allele with the -108 C, -162 A, and -909 G alleles [54]. Similarly, other investigators reported a correlation between the -107 C, -824 G, and -907 G genotypes and the higher PON1 serum concentration and activity towards both paraoxon and phenyl acetate. However, they suggested that predominant effect was exerted only by the -107 C allele occurrence and could not state any significant impact of the -824 and -907 positions [43]. Additionally, it was shown that the serum PON1 concentration is the highest in -108 CC homozygotes, medium in -108 CT heterozygotes and the lowest in -108 CC homozygotes. The -108 C/T SNP is localized in the binding site of Sp, ubiquitously occurring protein, taking part in the transcription of the TATA–less genes such as the PON1 gene [27]. Apart from that, the -162 polymorphic site may be localized in the region comprising both the binding site of NF-1 transcription factor [29] and the sequence similar to the IL-6 responsive element involved in the inflammatory process [55].

The detailed list of SNPs of the *PON1* gene was established due to the efforts of Nickerson and coworkers and is available at http://pga.gs.washington.edu (accessed on 3 December 2020). As a result of the PON1 gene resequencing process, the researchers found 180 new SNPs including eight new promoter region polymorphisms, one new coding region SNP, nine additional 3′ UTR SNPs and 162 additional intronic SNPs [56].

The analysis of the impact of *PON1* polymorphic sites assumed to be functional (192 Q/R, 55 L/M, -108 C/T and -162 A/G) on the PON1 activity allowed for only a partial explanation (about 25%) of the high rate of the PON1 variability between individuals. Aiming to evaluate the significance of all common polymorphic variations of both *PON1* and *PON2* in the PON1 AREase activity, Carlson et al. sequenced *PON1* DNA and analyzed taggingSNPs. The authors could state that regardless of the four already described functional SNPs, several previously unknown common *PON1* polymorphisms were also responsible for an additional 4% of PON1 arylesterase activity. The *PON1* gene was found as highly recombined [57]. Kim et al. added to the analytic model six novel intronic SNPs located in the *PON* gene cluster (rs854567, rs2299257, rs2237583, rs3917486 in *PON1*, rs2375005 in *PON2* and rs11768074 in *PON3*), making it possible to explain 30.1% of the PON1 arylesterase activity variation. Five of the six analyzed SNPs (excluding rs2237583) affected AREase activity, and three (including rs854567, rs2299257, and rs3917486) also exerteda significant influence on the paraoxon hydrolysis efficiency. The addition of as many as 51 evaluated SNPs to the regression model allowed explanation of84.96% activity variance toward paraoxon [31]. In the next study, the dependence of serum PON1 activity towards phenyl acetate and paraoxon from two SNPS in the intron 5 (17899insdelTT and 17974 CT) and the polymorphism 192 QR in the *PON1* gene coding region were evaluated. The PON1activity towards paraoxon was the highest in the presence of 17899insTT/insTT, 17974 CT or TT, and 192 RR genotypes, and the lowest for the 17899delTT/delTT, 17974CC, and 192 QQ genotypes. The significant LD was observed between all evaluated SNPs (the strongest between 17899insdelTT and 192 QR) and the multiple regression analysis revealed that from among genetic factors, the 17899insdelTT and 192 QR polymorphisms were significant determinants of PON1 activity towards paraoxon, while only the 192 QR genetic polymorphism was a significant determinant of the serum arylesterase activity. The authors suggest that the polymorphisms within introns may influence the serum PON1 activity as a result of the strict LD with the 192 QR polymorphism [58]. The *PON1* intronic polymorphisms probably affect the normal splicing of the gene’s mRNA and this way influence the regulation of the enzyme function [59]. A SNV localized in 3′ UTR of *PON1* and involved in a miRNA binding was also identified. Liu et al. reported that the presence of the T allele at rs373559 position results in the lesser miRNA-616 binding in this region and the *PON1* gene overexpression in comparison with the C allele [60].

There is also a suggestion that the *PON1* gene may be regulated by interchromosomal interactions. Winnier et al. reported the significant effect of one locus on chromosome 12 on PON1 levels [61] and suggested two other *PON*-related quantitative trait loci (QTLs) on chromosomes 17 and 19 [62]. Additionally, as was confirmed in multipoint analysis, *PON1* may be regulated by one of genes contained by chromosome 8 at p11, 21 [63]. The tightLD between promoter and coding regions 192 Q/R [64] and 55 M/L [54] variationswas reported. If the genotypes-108 CC at the promoter region and 55 LL at the coding region coexisted, the trend toward higher levels of the PON1 enzyme was noted [54]. In individuals in whom an inconsistency between *PON1* Q192R polymorphism and expected PON1 enzyme level was observed, the DNA sequencing allowed the researchers to identify a P90L SNV associated with the Q allele. Similarly an Asp124 missplice mutation associated with the R allele, as well as partial deletion of the *PON1* 5′ end, may affect PON1 status [65]. Furthermore, the effect of several polymorphisms of genes localized beside the *PON* gene cluster onPON1 activity regulation was explored and it was reported that both common and rare variants in genes such as *FTO, SERPINA12* and *ITGAL* may exert significant impact on the PON1 enzyme activity [66]

Moreover, it is known that hypermethylation in the region of cytosine-phosphate guanine (CpG) islands may lead to the transcriptional silencing of the gene expression [67]. It was also found that in the *PON1* promoter region, the higher DNA methylation level of CpG4 results in the lower expression of PON1 mRNA [68].

The PON2 protein, being the product of the oldest member gene of the PON family, unlike PON1 does not hydrolyze the OP, but it has properties of lactonase with the greatest hydrolysis rate of acyl-chloride in the PON family [18]. From two common polymorphisms in the coding region of *PON2* (147 A/G and 311 S/C), the functional role influencing the lactonase activity was established for the last one [69]. It should be mentioned that both considered polymorphic sites are tightly linked [70]. Recently, it was reported that SNP in the intronic part of *PON2* gene (rs2375005) also significantly affect PON1 AREase activity and the rate of paraoxon hydrolysis [31].

The *PON3* gene, situated between *PON1* and *PON2*, like *PON1* includes the TATA–less promoter region containing several polymorphic sites [2]. However, the studies concerning the *PON3*, being the most recently described gene from the *PON* family, are relatively scant. Similarly to PON2, PON3 act as lactonase and hydrolyses lipolactones, including statin lactones commonly used as substrates for PON3 in laboratory practice, and N-acyl–homoserine lactone, but have no activity towards OP [18]. The promoter variations -567 C/T, -665 A/G, -746 C/T were identified as significantly associated with PON3 concentration, which is the opposite of -4105 G/A, -4970 T/G, -4984 A/G, which are devoid of functional relevance. Additionally, the close LD of analyzed polymorphisms allowed the researchers to distinguish three various haplotypes: the less frequent TGTGTA, TGTAGG of medium frequency, and the most frequently occurring CACGTA [35]. The next studies evaluating these same six SNPs performed by the same authors in patients with HIV and with coronary artery disease (CAD) did not reveal any significant impact of the *PON3* genotype on the serum PON3 level [71,72]. The genotyping of 40 SNPs within the *PON* cluster gene reveals the difference in the hydrolysis rate of atorvastatin δ-lactone between a strongly linked group of *PON3* polymorphisms including -4984 A/G, -4105 G/A, -1091 A/G, -746 C/T, and F21F [73]. The arrangement of the *PON* cluster gene including polymorphic sites affecting enzyme activity and the sequences responsible for regulatory elements binding is depicted in Figure 1. The data summarizing the influence of genetic variations of the *PON* cluster noncoding DNA on PON activities are presented in Table 1.

## 4. *PON*s Polymorphism and Lipid Profile

The oxidation of low density protein(LDL) is the process involved in the development of atherosclerosis because the proinflammatory particles that arise during this process initiate the formation of atherosclerotic lesions [74]. In contrast, HDL plays an atheroprotective role, taking part in the reverse cholesterol transport and due to apolipoprotein A-I (ApoA-1) function, favoring the efflux of cholesterol from macrophages. In addition, HDL is capable to inhibit LDL oxidation [75]. Because both serum enzymes PON1 and PON3, which are bound to HDL, as well as PON2, which acts intracellularly, are able to potentiate macrophage cholesterol efflux and prevent lipoproteins oxidation [9,17], the issue of the relationship between *PON*s polymorphism and lipid profile was raised in many studies. However, the conclusions were not obvious [40]. Nevertheless, in patients with coronary heart disease(CHD) and age-matched controls, a highly significant positive correlation between both PON1 activities and concentrations and HDL-cholesterol and apoA-I level was found. The -107 C/T polymorphism in the *PON1* gene promoter was declared to be an independent predictor of HDL concentrations. In patients from the control group, if the genotype CC associated with the high expression of the *PON1* gene was present, the highest HDL concentrations were observed; this effect, however, did not occur in patients with CHD [76]. Furthermore, in Sicilian healthy octogenarians with significantly higher levels of HDL-C and PON1 activity compared with the controls the frequency of -107 CC genotypes was also significantly higher. In both groups the CC genotype was associated with higher PON1 activity and HDL levels than the low-expressor (TT) genotype [77]. 

It was presented that PON1 arylesterase activity was not significantly correlated with the susceptibility of LDL to the oxidation, opposite to genotypes in the promoter region such as -108 C/T (*PON1*1696) and a few other *PON1* SNPs including *PON1*13625, PON16054, PON112471, PON127678, *PON1*23887. The authors underlined the importance of the relationship found between *PON1* polymorphisms, in particular -108 T allele, and the LDL oxidation expressed as LDLmaxox. This relationship was independent ofthe PON1 activity, since after the removal of HDL from the LDL oxidation assay and after the inactivation of the calcium dependent active center byethylenediaminetetraacetic acid (EDTA), the presence of the functional PON1 enzyme was excluded. These findings allowed the authors to conclude that several *PON1* SNPs seem to be related to the LDL oxidative phenotype contributing to atherosclerosis [57]. Furthermore, in patients with diabetes significantly higher levels of oxidized LDL (Ox-LDL) in relation to apoB concentration (Ox-LDL/apoB ratio) were attributable to the -108 TT genotype associated with lower PON1 in comparison with the CT or CC genotypes [78]. Because it is believed that rather the number of PON1 particles than HDL concentration is responsible for the enzyme activity, the determination of the genetic background of the individual “oxidative status” may be a valuable additional factor in the predicting the risk of atherosclerosis, particularly in the case of normal lipid profile. The physiological extra- and intracellular actions of PON enzymes in the defense against the lipid oxidation are illustrated in Figure 2.

## 5. Clinical Relevance of *PON*s Polymorphism

### 5.1. Atherosclerosis–Related Diseases and PONs Polymorphism

Despite many years of intensive studies, the role of *PON* genes, especially noncoding DNA variations, in susceptibility to atherosclerosis, requires further exploration, because the results are not concordant. Because the essential physiological function of PON enzymes may be the hydrolysis of lipids peroxides and homocysteine thiolactones contributing to atherosclerosis development, it was assumed that individuals characterized the by genotype associated with the possible lower PON protein concentration and activity would be more prone to atherosclerosis. In the animal experiment, atherosclerotic lesions were less advanced in transgenic apoE null mice with the *PON* cluster than in nontransgenic apoE null mice fed with a high-fat diet, regardless of the lipid profile including total cholesterol, HDL cholesterol, VLDL/LDL cholesterol, triglycerides, or glucose levels. Nevertheless, the authors did not observe an increased effect of the entire gene cluster overexpression if compared with the *PON1* or *PON3* gene overexpression alone [79]. 

#### 5.1.1. Coronary Heart Disease

Aiming to evaluate which polymorphic forms of *PON* gene cluster may serve as predictors of coronary heart disease, several studies was conducted. Considering LD along the entire gene cluster, in majority of experiments the promoter and coding region variations were assessed simultaneously. Although it was found that the PON1192Q isoenzyme is more effective in hydrolyzing lipid peroxides in comparison with the 192R isoenzyme [80], some studies confirmed the relationship of the 192R genotype with CHD [81,82] while others did not report this effect [83,84]. The same contradictory results were reported regarding the 55 L/M polymorphism [84,85]. The newest meta-analysis, including one hundred and nine studies evaluating two missense mutations (*PON1*Q192R [rs662] and *PON1*M55L [rs854560]), revealed that in general population both evaluated polymorphic sites were significantly correlated with susceptibility to coronary heart disease and ischemic stroke, defined as atherosclerotic cardiovascular diseases. In Caucasians as well as East and South Asians rs662 polymorphism was found as a predictor of cardiovascular diseases, while rs854560 polymorphism plays similar role only in the East Asian subgroup. If only the association of evaluated polymorphisms with the coronary heart disease occurrence was analyzed, positive results for rs662 polymorphism were achieved [86].

The possible relationship between variations in *PON1* promoter region and coronary heart disease and a potential usefulness of these variations as a predicting factor remains unclear, although some studies seemed to be exceedingly promising. In the group of type 2 diabetic patients, the low-expressor genotype (TT) occurred significantly more frequently in patients with CHD. The observed association of the low-expressor genotype and the lower enzyme concentration with the increased risk of the disease was independent of other risk factors, including the coding region Q191R polymorphism. However, the authors underlined the coexistence of the promoter allele (-107 T) associated with the lower gene expression and the high-risk 191R allele and suggest a modulating effect of the promoter polymorphism on the risk of CHD [87]. In the study evaluating whether the promoter polymorphism -107 C/T of the *PON1* gene in the overall population is associated with the risk of CHD confirmed in coronary angiography, no association of the promoter genotypes and atherosclerotic lesion occurrence was found. However, the analysis of the subgroups revealed that the high expressor genotype (-107 CC) was connected with the lower risk of the disease in younger patients (≤60 years) independently of the other established risk factors for CHD [88]. The -907 C/G polymorphism in the promoter region was also evaluated as a possible risk factor for CHD. In the ECTIM2 population the GG genotype characterized by the higher gene expression was associated with a decreased risk of myocardial infarction and the greatest effect occurred in the subgroup of the youngest patients (<60 years) [89]. In the group of female patients with coronary heart disease and in the control group of healthy women genetic variations including the *PON1* promoter (-107 C/T, -162 G/A, -831 G/A) and coding region (160 R/G, 192 Q/R,) as well as *PON2* 311 S/C, and PON3 -133 C/A polymorphisms were evaluated. The authors showed significantly higher frequencies of -107 T and -831 G alleles in this group of patients compared with controls. Furthermore, the odds ratios for the CHD occurrence were 1.66 for -107 C/T heterozygotes and 2.0 for -107 CC homozygotes in comparison with TT homozygotes. The occurrence of haplotypes *PON1* -107 T and *PON2* 311S was significantly higher in cases compared with controls. The association of the mentioned haplotypes with CHD was significant after adjusting for environmental risk factors [90]. The question of the influence of the *PON1* genetic polymorphism within the 5′ region on the CHD severity was also investigated. In 99 patients and 66 controls undergoing coronarography the genotype and haplotype distribution of -907 G/C, -162 A/G and -107 C/T polymorphisms were different in patient subgroups with one-, two- and three-vessel disease. The haplotypes characterized by the high PON1 activity (-107 C and -907 G) were associated with a lesser extent of stenoses in the coronary arteries. However, the independent effect of analyzed genotypes on CHD severity was not confirmed [91]. Additionally, in patients with coronary heart disease undergoing coronary artery bypass-grafting (CABG) the-107 C/T polymorphism analysis allowed the researchers to predict the prognosis after operation. In the occurrence of the TT genotype the time of survival without serious cardiac events was significantly shorter compared with CT and CC genotypes [92]. Next, the analysis of the -909 G/C, -162 A/G and -108 C/T promoters as well as the192 Q/R and 55 L/M coding region polymorphismsas possible risk factors for CAD in northwest Indian Punjabis, an ethnic group characterized by a high incidence of CAD, revealed that the 192 RR and QR genotypes and the – 162 GG genotype were connected with CAD regardless of other conventional risk factors [93]. In contrast, no significant difference in genotype or allele frequencies at positions -107 C/T in the promoter and 192 Q/R and 55 L/M in the coding region of the *PON1* gene were found between patients with angiographically confirmed CHD and healthy controls [94]. Opposite to this, in the meta-analyses conducted by Wheeler et al. and by Wang et al., who evaluated results of, respectively,43 and 88 case-control studies regarding genetic variations at positions -108, 55 and 192 of the *PON1* gene and in position 311 of the *PON2* gene as potential risk factors of CHD, the investigators found only a weak relationship between the occurrence of the 192R *PON1* allele and CHD [95,96]. Furthermore, in the recently conducted large case control study involving 417 patients with CHD and 282 healthy individuals, no association between C-108T and G-909C promoter polymorphism and the CHD occurrence was observed. The significantly lower PON1 activity and concentration detected in the group of patients in comparison with the controls were independent of the genotype [97]. In the largest investigated population (~80,000 case-controls from the CARDIoGram consortium) Tang et. al. did not identify any association between the analyzed *PON1* SNPs, significantly affecting serum paraoxonase and arylesterase activity, and between prevalent CHD as well as MI history. In the prospective part of this study, after the three year follow up of a cohort involving 3668 subjects undergoing elective coronary angiography, no association of major adverse cardiac events (MACE) with the *PON1* polymorphism was found. Here, the authors considered the possibility that the study was underpowered as to genetic impact on prospective risk of MACE, since within the time frame of follow up, only 311 subjects from the angiographic cohort suffered MACE [20]. In the ARIC (Atherosclerosis Risk in Communities) population the occurrence of MACE including MI, stroke or CHD-related death and its dependence on race was evaluated, similarly to the possible association with *PON* cluster SNPs. From 82 SNPs available for the examined population, authors chose62, excluding these with minor allele frequencies < 5%. Unfortunately, none of SNPs that were included in the analysis fulfilled the baseline criteria of significance (the association with atherosclerosis in individuals of white and black race with the difference of *p* < 0.001) being, therefore, excluded from further examination. As a result no relationship between particular SNPs and MACE could be reported, despite the fact that this information would be very valuable in such a numerous cohort (12,770 participants) [98]. 

All above mentioned studies do not provide an unequivocal response about the role of promoter region SNPs as predictors of CHD. One possible explanation is that a remarkably variety of populations were assessed. The clinical relevance of the *PON* cluster gene polymorphism in coronary heart disease is summarized in Table 2. It should be noted that the association between the promoter region SNPs and CHD occurrence is more evident in populations more prone to atherosclerosis and was reported more frequently in smaller studies, whereas larger meta-analyses did not confirm these findings. The authors repeatedly underline the tight LD between coding and noncoding regions of the *PON* cluster gene. Undoubtedly, these findings indicate the need for large, prospective studies, conducted to a rigorous standard of accuracy, assessing the genetic background of CHD in relation to *PON* cluster gene variations. Nonetheless, genetic polymorphism evaluation may be a valuable tool for the evaluation of the risk of CHD in particular groups of patients (for example in a young subject without established classic risk factors of CHD experiencing acute coronary syndromes).

#### 5.1.2. Stroke

The genes involved in atherosclerosis development obviously seem to also be responsible for the occurrence of ischemic stroke (IS), particularly if resulting from atherosclerotic lesions of large arteries. Hence, the investigators are interested in the determination of the role of *PON* gene cluster in stroke, involving, however, not only atherosclerosis of the large arteries but also small vessel occlusion, cryptogenic or even hemorrhagic stroke. Most of the studies have focused on coding region variations and, like in the case of CHD, the results are contradictory and dependent on the examined population [99,100,101], which is contributed to ethnic heterogeneity. In a meta-analysis assessing rs662 polymorphism and including 22 eligible studies with no evidence of between-study heterogeneity (total of 7384 cases/11,074 controls), only the modest elevation in the risk of IS associated with the G allele of *PON1* rs662 was revealed [102]. In contrast, a more recent meta-analysis proved that *PON1* rs854560 polymorphism may be useful in predicting increased susceptibility to IS [86]. The studies assessing an association of noncoding DNA and stroke arescarce. In the group of young patients (age < 45 years) with first cryptogenic arterial IS, besides the Q192R and the L55M coding region polymorphisms, two polymorphic sites—107 C/T and -824 G/A in the promoter of the *PON1* gene—were also investigated. The authors reported that the occurrence of -107 T allele independently increased the overall risk of IS and additionally in the presence of 192 RR genotype, the -107 T allele contributed to a generally (although not significantly) higher risk of stroke in comparison with individuals without the T allele. This may suggest that -107 T allele interacts and enhances the risk associated with Q192R polymorphism [103]. On the other hand, it was reported that the -107 TT genotype almost doubled the risk of stroke in the elderly group of 172 unrelated adult Caucasian patients with acute hemispheric IS (age > 59). Additional findings indicated the protective role of the triple combined haplotype of assessed SNPs (-107 C/T, 55 LM and 192 QR) as well in the total as in the elderly population. The increased risk of IS was related to the 55 LL genotype, but PON1 paraoxonase, arylesterase and diazoxonase activities in the patients and the controls were comparable. It allowed the conclusion that the analyzed *PON1* genotypes, but not its activities, appeared to be related with the risk of IS [104]. Likewise, in patients with established atherosclerosis risk factors, an association between *PON1* genotype and early forms of atherosclerosis of carotid arteries were examined. In 187 patients with familial hypercholesterolemia, amongst the seven most commonly analyzed SNPs involving 55 L/M, 192 Q/R, in the coding region and -107 C/T, -126 G/C, -162 A/G, -824 A/G, and -907 G/C in the promoter region, only -824 A/G appeared to be an independent predictor of the carotid intima–media thickness (IMT). If the impact of particular haplotypes on the carotid IMT was assessed, the difference in IMT between theC-907/G-824/G-162/C-126/T-107/55M and-907 G/-824 A/-162 A/-126 G/-107 C/L55 haplotypes amounted to about 22%. [105]. Kim et al. examined only SNPs in the *PON1* 5′-regulatory region and serum PON1 activity as possible risk factors of IS resulting from the small vessel occlusion. From amongst the investigated 10 SNPs only the-1434 GG genotype was less frequent in patients, although the data did not achieve statistical significance (*p* = 0.06) [106]. In the large case–control study evaluating the risk of ischemic and hemorrhagic stroke dependent on SNPs in the Han Chinese population, in the entire *PON* cluster, eleven SNPs were genotyped. There were rs662 (Gln192Arg), rs13306698 (Arg160Gly), rs854560 (Leu55Met) in the PON1coding region; rs705379 (-107/-108), rs705381 (-160/-162), rs854571 (-824/-832), rs854572 (-907/-909) in the *PON1* promoter region; rs12026 (Ala148Gly) and rs7493 (Ser311Cys) in the *PON2* coding region, as well as rs2074353 (intronic SNP) and rs1053275 (Ala99Ala) in the *PON3* coding region. If the association of particular SNPs to the total stroke risk was evaluated, any significant difference was not found. However, after adjustment for any confounders such as age, HDL and hypertension, the polymorphism at position rs705381 was significantly associated with ischemic stroke. In their analysis based on identified LD, the authors distinguished four haplotype blocks (1 including rs854560, rs13306698 and rs662; 2: rs854572, rs854571 and rs705381; 3: rs1053275 and rs2074353 and 4: rs12026 and rs7493) and investigated the association between the SNPs included in particular blocks and the risk of stroke using dominant, recessive, additive and genotype models. In a dominant model, the rs705381 (-162) polymorphism appeared significantly associated with the total risk of stroke as well as the risk of ischemic stroke, but there was no association with hemorrhagic stroke. Regarding the nominal association that has been found between rs854571 (-824) and stroke, the allele T was identified as a protective factor. The researchers suggest the possible role of the transcriptional activator NF-1, binding at the -162 position of *PON1* promoter region in the presence of allele A [107]. These promising results encouraged other investigators to undertake further studies. Six genetic polymorphisms of *PON1* and *PON2* genes earlier reported as associated with IS risk localized in exons (*PON1* 192 Q/R +16342 C/T, *PON2* C311S +34610 C/G), promoter regions (-162 A/G, -832 C/T, -909 G/C) and 3′UTRs (+26080 T/C) were examined in a cohort of 300 Chinese patients with diagnosed IS and 300 healthy controls. The authors identified only the *PON1* rs662 R(G) allele as a potential risk factor for IS. The impact of this allele on the risk of stroke was particularly apparent in the male subcohort in comparison with females. In addition, the risk of IS associated with *PON1* rs662 polymorphism was more evident in patients with large-artery atherosclerosis [108]. Liu et al., on the basis of the finding that SNPs due the alteration of the microRNA (miRNA) binding sites can influence the genes expression, investigated the miR-SNPs potentially related to the risk of stroke. From among 75 genes regarded as stroke–responsible, the miR-SNPs were detected in 13 and, finally, the miR-SNP rs3735590 at the 3′ UTR of *PON1* gene was identified by using the miRNA arrays. Further evaluation revealed that in patients with the CT or TT genotype at rs3735590, the risk of IS was lower than in patients with the CC genotype. The risk of subclinical atherosclerosis associated with CC genotypes was also elevated. In the healthy controls with the CT or CT genotypes, atherosclerotic lesions of internal carotid arteries confirmed ultrasonographically were thinner than in individuals with the CC genotype. By using plasmid constructs containing the unfavorable allele C, the authors proved that *PON1* is a direct target gene for miR-616. The presence of the C allele at rs3735590 enhanced the miR -616 binding ability and inhibited *PON1* expression. On the contrary, the occurrence of the T allele suppressed miR-616 binding, which resulted in PON1 gene overexpression and possible protection against atherosclerosis and stroke [60]. The next study investigating the relationship between the methylation level of *PON* gene cluster and the risk of cerebral infarction fit the general trend of exploring epigenetic factors influencing the expression of genes. Here, in the studied group consisting of 152 patients with confirmed cerebral infarction, the *PON1* methylation level was significantly higher than in the 152 healthy individuals who served as the control group. Conversely, *PON3* methylation level appeared significantly lower in the case group compared to the control group. In performed subgroup analysis, an association of *PON1* and *PON3* methylation levels with stroke was found only in males. The higher *PON3* promoter region methylation was identified as a protective factor for cerebral infarction together with HDL and uric acid levels [109]. The clinical relevance of thenoncoding DNA polymorphism of the *PON* cluster in stroke is summarized in Table 3.

#### 5.1.3. Diabetes

It is well documented that diabetes increases the risk of atherosclerosis development and facilitates atherosclerotic complications [110], so the possible relationship between PON gene cluster polymorphisms and diabetes became a subject of current interest. However, the results of the published studies are not univocal. In a recently performed meta-analysis regarding *PON1* coding region polymorphisms (Q192R, rs662 and L55M, rs854560) and the susceptibility to type 2 diabetes mellitus in different ethnic populations, it was found that the 192R allele presents a protective role in Europeans, while in South and East Asians is a risk factor for diabetes mellitus (DM). The 55M allele was found to be a protective factor in European population and exerted no effect in Asians [111]. Regarding *PON1* promoter region polymorphisms, it should be emphasized that diabetic patients were the first group in which the relationship of -107 T allele of *PON1* and CHD occurrence was identified. The TT genotype, associated with significantly decreased PON1 concentrations, occurred more frequently in type 2 diabetic patients with CHD, predicted the risk of vascular disease independently of other risk factors and modified the effect imposed by the high risk allele 192R [87,88]. Since then numerous studies analyzing the relationship between *PON1* promoter polymorphism and diabetes mellitus in different populations were conducted. North Iranian patients suffering from type 2 diabetes carried the TT and CT genotypes significantly more commonly than healthy controls, while the favorable CC genotype was significantly less frequent in this group of patients in comparison with the healthy control group [112]. Similarly, in northwest Indian Punjabis, a distinct ethnic group characterized by the high incidence of type 2 diabetes mellitus(T2DM), after analyzing Q192R, L55M polymorphisms in the coding and -909 G/C, -162 A/G, -108 C/T in the promoter region of the *PON1* gene, the association of the QR, RR (Q192R) and GC, CC (-909 G/C) genotypes and the L-C-A-R-G, L-T-A-R-G, L-T-G-Q-C haplotypes with T2DM was detected. In diabetic patients the PON1 enzyme activity towards paraoxon and phenyl acetate was significantly lower than in the control group [113]. In the European population the relationship between *PON1* promoter polymorphisms and diabetes was not obvious. It was reported that in type 1 and type 2 diabetic patients only L55M and Q192R polymorphisms in the coding region, but not -107 C/T and -907 G/C in *PON1* promoter region were significantly associated with the disease occurrence (allele M is more frequent and allele R is less frequent in the group of patients than in the control group as well as in patients with macroangiopathy versus patients without it). Genotypes LL and RR favorably modified the diabetes control and were connected with the higher PON1 enzyme activity [114]. In the study conducted by our group, the difference in the -107 C/T genotypes distribution in the group of patients with overt CHD and diabetes in comparison with normoglycemic CHD patients appeared insignificant, but PON1 paraoxonase activity—known to be closely related to the genotype—was found as a factor independently associated with diabetes [22]. The high glucose level seems to be one of factors modifying the *PON1* gene expression due to interaction between Sp1 and protein kinase C (PKC). In cultured human hepatocyte cells Huh7 with transducted *PON1* (-1232/-6)-luciferase expression vector cells glucose infusion induced *PON1* promoter activity in the dose-dependent manner, which resulted in the pronounced transcription and expression of *PON1*. The inhibition of PKC by bisindolylmaleimide significantly suppressed the glucose-dependent activation of the *PON1* promoter, and the using an inhibitor of Sp1 mitramycin completely stopped the *PON1* transactivation. The authors suggest that the observed phenomenon seems to be a compensatory mechanism in diabetes, a disease characterized by the depletion of antioxidant potential, including PON1 enzyme activity [115]. The influence of genetic variations on PON1 enzyme activity in diabetes is revealed as early as in neonates of women suffering from gestational diabetes mellitus. Recently it was found that in neonates of mothers with GDM with abnormal fasting glucose, the *PON1*–108 T allele was associated with decreased lactonase activity in comparison with the neonates of normoglycemic mothers [116]. Furthermore, innondiabetic patients the *PON1* promoter polymorphism -107 C/T was identified as an independent marker for abnormal fasting glucose concentrations. In patients withabnormal glucose control a higher frequency of TT and CT genotypes—associated with lower expression of the *PON1* gene—was found. The authors explained their findings by a possible predisposition of individuals with low-expressor genotypes to insulin resistance or the linkage of *PON1* promoter region variations with another gene involved in glucose control [117]. The association of *PON1* genetic polymorphisms with insulin sensitivity estimated by the euglycemic hyperinsulinemic clamp was evaluated in Japanese patients with type 2 diabetes. The authors noted that the increase in glucose infusion rate (GIR), was the lowest for -108 TT, intermediate for -108 CT and the highest for the -108 CC genotype. The observed differences were statically significant and allowed the authors to conclude that -108 C/T *PON1* promoter region polymorphism influences insulin sensitivity [118]. The most unfavorable effects of diabetes are macro- and microvascular complications. It was shown that *PON1* coding region and promoter variations (which remain in the strong LD) were significantly associated with diabetic complications. In patients with T2DM, *PON1* alleles 54L, -162 A and -1074 G, which lead to the lower PON1 activity, were significantly associated with the higher urinary albumin loss indicating the diabetic nephropathy. The genotype -907 GG was identified as a protective factor against retinopathy [119]. As was exhibited in Japanese patients with T2DM, the genotype TT at the -108 C/T polymorphic site in the promoter region of the *PON1* gene, which is associated with decreased *PON1* expression, contributed to the unfavorable, proatherogenic lipid profile. In individuals with the TT genotype the Ox-LDL/apoB ratio was higher than in the case of CT or CC genotypes and it was associated with the presence of macrovascular disease and peripheral neuropathy [78]. On the contrary, in patients with type 1 (insulin-dependent) diabetes mellitus and diabetic nephropathy any differences in genotype and allele frequencies for three common *PON1* polymorphisms (-107 C/T, 54 L/M and 192 QR) were not found in comparison with diabetic patients without nephropathy [120]. The clinical relevance of the noncoding DNA polymorphism of the *PON* cluster in diabetes is summarized in Table 4.

#### 5.1.4. Peripheral Artery Disease, Dementia and Renal Disease

The data concerning the relationship of *PON* gene cluster with other atherosclerosis-related diseases involving peripheral artery disease (PAD), nephropathy or dementia are considerably less numerous. PAD is defined as an abnormal narrowing of arteries providing blood supply to the limbs and abdomen and it results from atherosclerosis. Although increased levels of PON1 and PON3 were immunohistochemically identified in the endothelium of arteries obtained from patients with serious PAD during the surgical infra-inguinal revascularization [121] and significantly higher PON1 and PON3 serum activity, as well as positive correlations between PON3, insulin levels and insulin resistance were found in patients with PAD [72], there is no convincing evidence for genetic causes of these findings. The effect of *PON1* on the vasodilatation was also investigated, because endothelial dysfunction is regarded as the earliest stage of atherosclerosis. In patients with PAD the *PON1*192R allele occurrence was associated with the increased PON1 activity and the better function of endothelium presented as brachial artery flow-mediated vasodilation in comparison with 192Q allele carriers. However, only the PON1 serum activity, not the genotype appeared to be the independent predictor of worse endothelium function [122].

Renal disease is known as a factor considerably accelerating the progress of atherosclerosis. On the other hand, established risk factors for atherosclerosis, such as hypertension or diabetes, significantly impair renal function, contributing to overt renal insufficiency. In the group of 263 hemodialyzed patients Q192R, L55M, -108 C/T variations of the *PON1* gene and S311C of the *PON2* gene were evaluated. Only the frequency of the *PON1* 55MM genotype, as well as PON1 paraoxonase activity, were lower in hemodialyzed patients in comparison with the control group. The authors suggest that other factors beyond the genetic variability may be the cause of decreased PON1 activity in patients with severe renal insufficiency requiring hemodialyses [123]. In patients with systemic lupus erythematosus (SLE) the risk of lupus nephritis was associated with three SNPs in the *PON1* promoter region (-1074 A/G, -160 G/A, and -107 C/T). Significantly higher frequencies of -160 A, -1074 G, -107 C alleles were observed in patients with renal disease in comparison with the group of patients with the unaffected renal function [124]. Furthermore, in diabetic nephropathy alleles -162 A and -1074 G as well as the 54 LL genotype, all associated with the lower PON1 activity, contributed to significantly higher urinary albumin loss [119]. In the study evaluating the impact of the *PON1* gene promoter (-108 C/T and -162 A/G) and coding regions (Q192R and L55M) polymorphisms on IgA nephropathy occurrence and progression, the authors did not find any differences in the analyzed genotypes’ frequency between the group of patients and the control group as well as between patients with the nonprogressive versus progressive form of disease. However, the frequency of the -108 C allele and PON1 activity were significantly higher in patients with the nonprogressive disease than in individuals with the progressive nephropathy [125].

Genetic determinants including *PON* gene cluster polymorphisms were investigated in dementia of vascular origin (VD), Alzheimer’s disease (AD) and mixed types of dementia (MD). In 304 patients with all types of dementia the *PON1*-108T allele occurred significantly more frequently in patients with AD than in controls, but the prevalence of Q192R genotypes was similar in both investigated groups. Similarly, both polymorphisms affected PON1 arylesterase activity [126]. Since the abnormal cholesterol metabolism is regarded as involved in AD development, in patients with AD of late-onset type and in healthy elderly controls not taking any lipid-lowering drugs and with no cerebrovascular lesions in neuroimaging, the relationship between the plasma lipoprotein profile and LDL density and between *PON1* promoter genotypes was assessed. In the plasma of patients with AD and -107 TT genotypes smaller, denser LDL, as well as an elevated level of oxidized LDL were observed more frequently. The authors explain the altered lipid profile by the ineffective antioxidant plasma ability in carriers of -107 TT genotype, what results in the progression of vascular endothelium injury [127]. In the small population of patients with Alzheimer’s disease the T allele was more frequent regardless of age and gender compared with healthy controls, in whom the C allele was predominant [128]. In the larger group of 730 Caucasian and 467 African American subjects with AD enrolled in the MIRAGE study, 29 SNPs in the *PON1, PON2* and *PON3* genes were analyzed, including 192 Q/R, 55 L/M in the coding region of *PON1* and 311 C/S of *PON2*, 31 F/S of *PON3*, -107 C/T and -161 C/T in the promoter region of the *PON1* gene as well as intronic and intragenic variations. The most significant association with AD in both investigated ethnic groups was found for the -161 C/T polymorphic site of the *PON1* gene located at a potential binding site of the transcription factor NF-I [129]. Attempting to reproduce these findings, in a large study of patients with the sporadic form of AD and controls, French investigators confirmed functional variants related to AD development in the proximal promoter and 5′sequence of *PON1* [130]. In the further analysis of the MIRAGE population, the relationship of the *PON* cluster gene SNPs with the paraoxonase activity towards phenyl acetate and thiobutyl butyrolactone with AD was assessed. It was found that not only is decreased serum paraoxonase activity a risk factor for AD, but also that numerous DNA sequence variations in different regions of the *PON* gene cluster affect the serum paraoxonase activity more than polymorphisms within a single gene. The authors suggested the synergetic role of *PON1, PON2* and *PON3* genes in AD development [131]. In contrast, in apopulation of 306 Italian patients with AD and 275 controls only the slightly, non-significantly higher prevalence of the T allele was noted in AD patients in comparison with the controls, but no association between the *PON1* polymorphism and AD was found [132]. The DNA methylation is also regarded as a process involved in the development of dementia. In the group of patients with dementia of different types (53 with Alzheimer’s disease, 17 with vascular dementia and 32 with mixed dementia) a significant association was reported of the global DNA methylation and markers of the folate status with variations of genes previously reported as involved in dementia (among them *PON1* 192 Q/R polymorphism) [133]. The results of studies investigating the impact of PON gene variants on AD were summarized with the using ameta-analysis. The authors noted that from amongst five studied polymorphisms only the *PON2* 311SS genotype and the *PON1* rs705379 allele A were associated with the higher risk of AD, and the GG genotype of *PON1* rs705379 played a protective role. The -161 C/T polymorphism of *PON1* as well as 55 L/M and 192 Q/R did not influence AD occurrence [134]. The clinical relevance of noncoding DNA variations of the *PON* gene cluster in peripheral artery disease, dementia and renal disease is summarized in Table 5.

Although several *PON* cluster gene SNPs were examined as possible predictors of atheroscerosis development, finally only a few appeared involved in atherosclerosis-related diseases, as depicted in Figure 3. 

### 5.2. The Response to Drugs

#### 5.2.1. Antiplatelet Drugs

In the treatment of all clinical forms of atherosclerosis and in prevention of cardiovascular events, especially in patients after percutaneous vascular interventions, a key role is played by antiplatelet drugs – aspirin and inhibitors of platelet receptor P2Y12 such as clopidogrel and, increasingly, by newer drugs such as tikagrelor and prasugrel. The response to antiplatelet therapy significantly differs between particular individuals and some patients suffer from cardiovascular events despite the applied treatment. The possible mechanism of this phenomenon seems to be antiplatelet drug resistance resulting from various factors [135]. It was suggested that *PON1* is an important enzyme for clopidogrel bioactivation, being involved in the transformation of 2-oxo-clopidogrel into the active substance, which was confirmed by in vitro experiments and assessed ex vivo by ADP- induced platelet aggregation. Based on the investigation of patients who survived a stent thrombosis despite clopidogrel therapy, a hypothesis that *PON1* Q192R polymorphism is associated with the risk of cardiovascular events on clopidogrel therapy was presented [136]. The authors of this study found that *PON1* 192 Q/R polymorphism is responsible for over 70% of the variability in platelet aggregation and consequently patients with the *PON1* 192 QQ genotype were more prone to the stent thrombosis and presented both lower PON1 plasma concentration and activity as well as decreased platelet inhibition. Nevertheless, none of the subsequently conducted studies in patients undergoing elective or urgent percutaneous interventions demonstrated the relationship between the polymorphism in thePON1 coding region and the risk of cardiovascular events [137,138,139]. This same finding was revealed in a meta-analysis summarizing the role of *PON1* Q192R polymorphism in the biological response to clopidogrel for the risk of MACE [140]. This fact may be explained by an experiment in which the reaction of the clopidogrel active metabolite formation, if catalyzed by an esterase such as PON1, gives as a product a transdiastereosomer, which is less efficient in inhibiting platelet aggregation than a cisactive metabolite[141].However, amore recent study reports that in patients with CHD undergoing percutaneous coronary intervention, who were treated with aspirin and clopidogrel and followed for a year, the 192 RR genotype of *PON1* occurred more frequently in clopidogrel-resistant patients and was associated with clopidogrel resistance evaluated as the ADP-induced platelet aggregation rate [142].The risk of high platelet responsiveness (HPR) estimated by thrombelastography was higher in patients with acute coronary syndrome and the 192 RR genotype of *PON1* [143]. Because during treatment with antiplatelet agents the balance between the risk of thrombosis resulting in recurrent cardiovascular events and the risk of bleeding is important, the impact of *PON1* genetic polymorphism on the clinical outcome in the form of MACE or bleeding occurrence was evaluated in patients with CHD after percutaneus coronary intervention PCI. In individuals with the-126 C allele of *PON1* gene the presence of CHD as well as the risk of bleeding within six months after PCI were increased in comparison with individuals without this allele. Additionally, it was observed that the occurrence of the haplotype p-162A-126C-108C in the *PON1* promoter region resulted in a gene transcription that was significantly higher when compared with the most frequent haplotype p-162G-126G-108T [144]. The relationship between DNA methylation depending on genetic variations in the PON1 gene promoter and between bleeding or MACE occurrence during dual antiplatelet therapy was also evaluated in patients with CAD, who underwent PCI. For all five *PON1* CpG sites analyzed (-142, -184, -170 -163 and -161) a weak but significant association between the lower methylation level and bleeding events was confirmed. For the CpG site -163 aspirin reaction units (ARUs) also appeared to be significantly associated with the DNA methylation level. In carriers of the *PON1* -126 CC genotype ARUs was significantly higher in comparison with individuals carrying the -126 CG genotype. Additionally, PON1 methylation level at CpG site -161 was reported as indicating a lower risk of bleeding [145]. However, in another study including patients with dyslipidemia, the higher DNA methylation level of four CpG dinucleotides in the PON1 promoter region was associated with poorer clopidogrel response, which according to authors resulted from the lower expression of PON1 mRNA [68].

#### 5.2.2. Statins

The second group of drugs widely administered in atherosclerosis-related diseases are statins, these being 3-hydroxy-3-methyl-glutaryl-CoA reductase inhibitors. These agents, besides their lipid lowering properties, exert wide pleiotropic actions by influencing the endothelium function, stabilizing atherosclerotic plaques, inhibiting coagulation, stimulating fibrinolysis, inhibiting inflammatory reactions and acting as immunomodulators. The impact of statins on the cardiovascular system also involves the increase inPON1 activity resulting from various molecular mechanisms. In vitro, in expression cassettes transfected to HepG2 cells, it was shown that simvastatin enhanced *PON1* promoter expression via nuclear factors in a dose-dependent manner, in particular regarding SRE- binding protein-2 located in the *PON1* promoter region. Furthermore, in patients treated with simvastatin the concentration and activities of PON1 significantly increased. Because the *PON1* promoter fragment implies the function of mentioned nuclear factors involves the -107 C/T polymorphic site, it was investigated in vitro as well as in vivo, how the -107 C/T genotype impacts statin action. It was observed that the stimulatory effect of simvastatin was more potent in the case of the C than of the T allele occurrence. The expression increase, which results from SREBP-2 binding, was significantly stronger for the C promoter. Furthermore, in patients with hypercholesterolemia treated with simvastatin a significant increase in PON1 activity and mass occurred in the presence of CC allele, while in patients homozygous for the T allele such an increase was not found [45,146]. The activating effect of pitavastatin on the PON1 promoter and the increased PON1 protein expression was also shown in cultured Huh7 cells. Pitavastatin—by the phosphorylation of p44/42 mitogen activated protein (MAP) kinase—stimulated the PON1 promoter as well as the PON1 protein expression. The experiment revealed that the effect exerted by pitavastatin resulted from Sp1 binding to the *PON1* promoter and from the phosphorylation of SREBP-2, because inhibitors of Sp1, such as mitramycin, suppressed pitavastatin-dependent *PON1* promoter activity and a transfection of investigated cells with SREBP-2 deprived of normal p44/42 MAP kinase phosporylation sites decreased *PON1* promoter activity [147]. The stimulation of the Sp1 binding to the *PON1* gene upstream by pitavastatin were observed also in Huh7 cells. The effect of this statin on *PON1* promoter activation and protein expression depended on atypical protein kinase C (PKC) activation [148]. Another suggested mechanism of pitavastatin-induced *PON1* promoter activation was the farnesyl pyrophosphate (FFP) pathway observed in the plasmid model transduced into human hepatoma HepG2 cells and human embryonic kidney (HEK) cells [149]. Nevertheless, it should be mentioned that in the current clinical practice, the most powerful agents, such as atorvastain and rosuvastatin, are regularly used. The activating impact of atorvastatin on *PON1* promoter region was reported [146]. The clinical evaluation of *PON1* genetic variations’ (involving192 Q/R, 55 L/M and -107 C/C polymorphisms) influence on the effectiveness of atorvastatin administration was performed in Sicilian subjects with hypercholesterolemia. After three weeks of treatment with 10 mg of atorvastatin, favorable differences the in lipid profile as well as improved resistance of LDL to oxidation and increased PON1 activity were observed regardless of genotype [150]. The molecular mechanisms of the statins’ impact on *PON1* gene expression are depicted in Figure 4. Based on known PONs lactonase properties, the ability to hydrolyze the toxic, pharmacologically inactive metabolite atorvastatin δ-lactone by different genetic forms of *PON1* and *PON3* was investigated in the human liver bank. Forty polymorphisms along the *PON*-cluster were genotyped and it was shown that SNPs -108 T/C, -832 G/A, -1741 G/A of the *PON1* promoter and strongly linked SNPs -4984 A/G, -4105 G/A, -1091 A/G, -746 C/T, and F21F of *PON3* were associated with differences in atorvastatin δ-lactone inactivation. Seven functionally important haplotypes were distinguished and less frequent haplotypes consisting of PON3 SNPs or PON1 promoter SNPs were described as more efficiently hydrolyzing δ-lactone in comparison with the most common haplotype including all other functional haplotypes. These findings, according to authors, may suggest a genetic background for adverse effects of statin therapy [73].

## 6. Conclusions

The noncoding DNA variations of *PON* gene cluster exert a considerable effect on the occurrence and the course of atherosclerosis-related diseases.The estimation of the predictive role of the *PON* gene polymorphism in the general population requires large prospective studies with appropriate accuracy of conduct. The data achieved until now allow us to consider *PON* cluster gene variations as additional atherosclerosis risk factors in selected individuals (for example in young patients without conventional risk factors who experienced cardiovascular events). Furthermore, in populations particularly prone to atherosclerosis, such as diabetic patients, these polymorphisms may serve as additional predictors of the risk of atherosclerosis.The issue of paramount importance that should be studied is the relationship between noncoding DNA *PON* genes variations and the drugs’ response, especially the clopidogrel resistance and adverse effects of statins occurrence, because the identification of compromised patients’ possible complications may enable the tailored therapy administration.

## Figures and Tables

**Figure 1 ijms-22-02137-f001:**
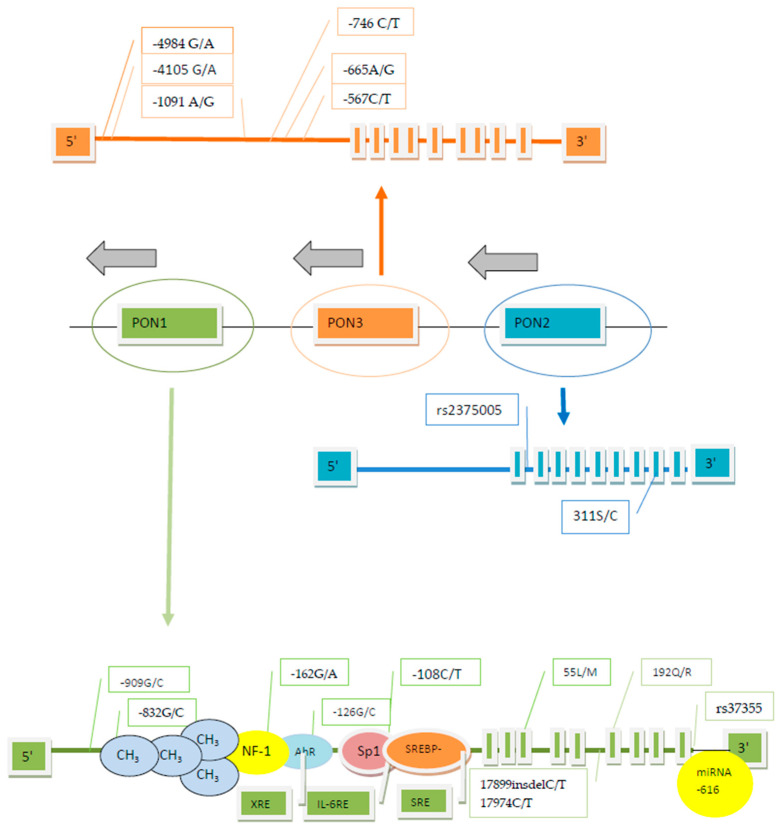
The arrangement of the paraoxonase (PON) cluster gene including polymorphic sites affecting enzyme activity and sequences responsible for regulatory element binding (according to [35,41,42,43,44,45,46,47,54,57,58,60,68,73]).

**Figure 2 ijms-22-02137-f002:**
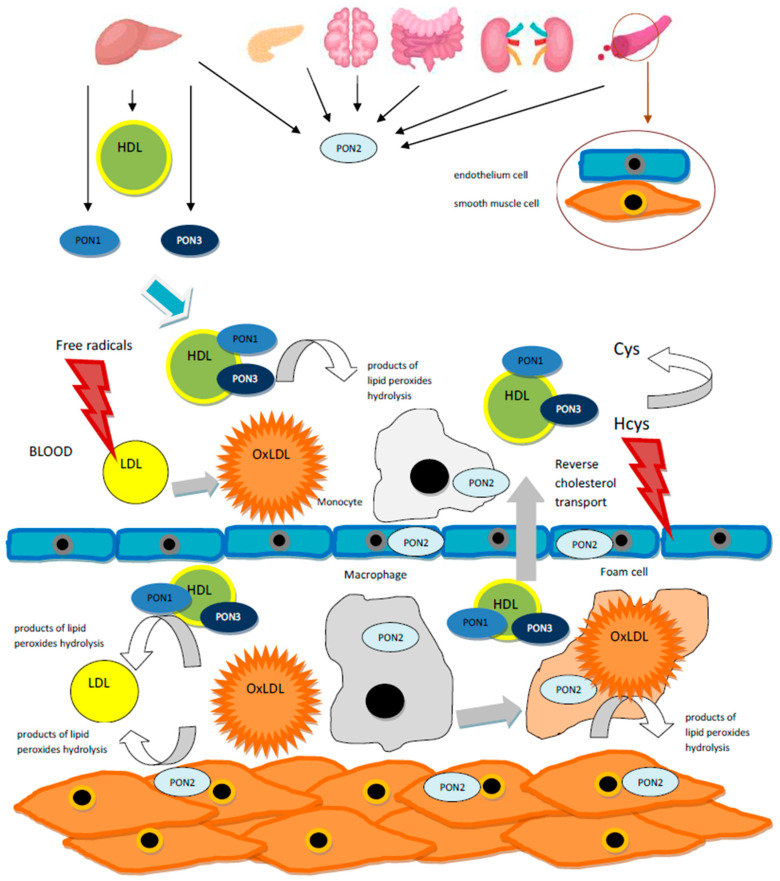
The physiological actions of PON enzymes in the defense against lipid oxidation. PON1 and PON3 proteins are produced in the liver and excreted into plasma where they are bound to HDL.The PON2 protein is found intracellularly in the liver, brain, kidneys, heart, pancreas, intestine and endothelial and smooth muscle cells. All three enzymes are able to potentiate macrophage cholesterol efflux and to prevent lipoprotein oxidation by hydrolyzing lipid peroxides and, due to lactonase activity, to neutralize the homocysteine (Hcy) thiolactone tissue toxicity [7,8,9,15,17].

**Figure 3 ijms-22-02137-f003:**
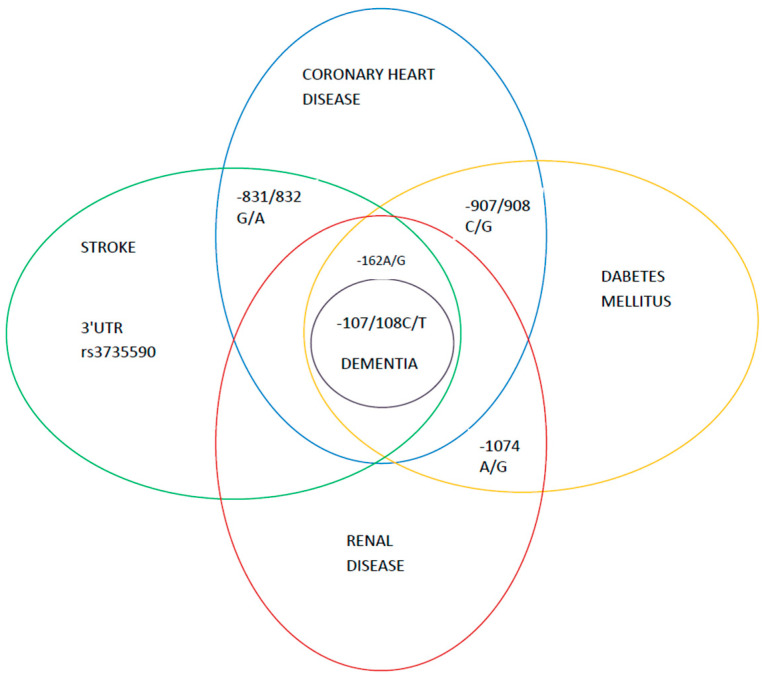
The *PON* cluster gene SNPs involved in the atherosclerosis-related diseases. The central role of the -108 C/T polymorphisms.

**Figure 4 ijms-22-02137-f004:**
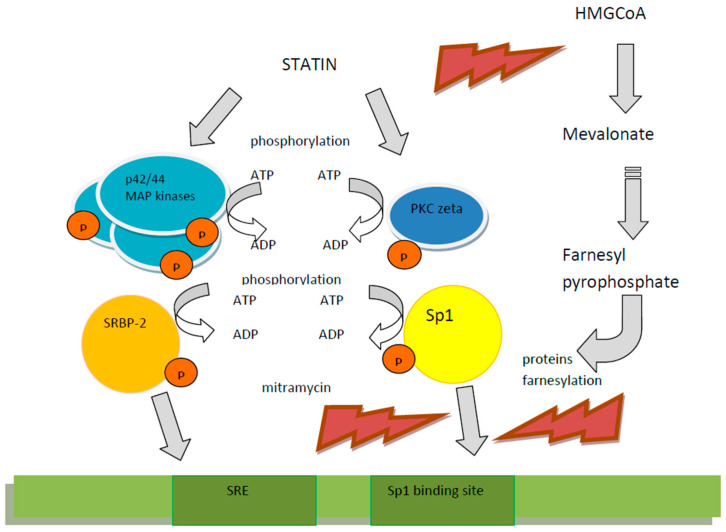
The molecular mechanisms of statins’ impact on PON1 gene expression in the example of pitavastatin. Pitavastatin increases phosphorylation of p44/42 mitogen activated protein (MAP) kinase and protein kinase C stimulates Sp1 and SREBP binding to the PON1 promoter and enhances the gene expression [147,148]. The inhibition of the farnesyl pyrophosphate formation results in enhanced binding of regulatory proteins to the PON1 promoter [149].

**Table 1 ijms-22-02137-t001:** The relationship of noncoding DNA of *PON* cluster gene variations and PON’s activity.

PON Activity	Polymorphic Site (Functional Alias)	DbSNP Rs Number	References
Responsible for 22.8% variation in arylesterase activity; C allele associated with higher PON1 activity towards paraoxon, phenyl acetate and PON1 concentration; the highest PON1 activity and concentration is seen in CC genotype, medium in CT and the lowest in low-expressor genotype (TT)	-107/108 C/T; *PON1* promoter	rs705379	[31,43,54,57]
Influence on PON1 arylesterase activity	-162 A/G *PON1* promoter	rs705381	[54]
Influence on PON1 arylesterase activity	-907/909 C/G *PON1* promoter	rs854572	[54]
Influence on PON1 arylesterase and paraoxonase activity	*PON1* introns	rs854567	[31]
		rs2299257	
		rs3917486	
	*PON2* intron	rs2375005	
PON1 activity towards paraoxon higher in the presence 17899insTT/insTT and 17974 CT or TT PON1 activity towards phenyl acetate higher for 1799delTT/del TT	17899insdelC/T *PON1* intron 5 17974 C/T *PON1* intron 5		[58]
Associated with PON3 concentration	-567 C/T *PON3* promoter -665 A/G *PON3* promoter -746 C/T *PON3* promoter		[35]
Differences in PON1 efficiency in atorvastatin δ -lactone hydrolysis	-4984 G/A *PON3* promoter -4105 G/A *PON3* promoter -1091 A/G *PON3* promoter -746 C/T *PON3* promoter		[73]

**Table 2 ijms-22-02137-t002:** The clinical relevance of noncoding DNA polymorphism of *PON* cluster in coronary heart disease.

Clinical Relevance	Polymorphic Site (Functional Alias	DbSNP Rs Number	References
TT genotype more frequent in diabetic patients with CHD; CC genotype associated with lower risk CHD < 60 years old; T allele more frequent in women with CHD; C allele associated with less severe coronary artery stenoses; shorter cardiac events free survival after artery bypass—grafting (CABG) in TT and CT genotypes	-107/108 C/T; *PON1* promoter	rs705379	[87,88,90,91,92]
GG genotype independently associated with coronary artery disease (CAD) in northwest Indian Punjabi population	-162 A/G *PON1* promoter	rs705381	[93]
GG genotype associated with decreased risk of myocardial infarction 60 years old; and less severe coronary artery stenoses	-907/909 C/G *PON1* promoter	rs854572	[89,91]
G allele more frequent in women with CHD	-831 G/A *PON1* promoter	rs854571	[90]

**Table 3 ijms-22-02137-t003:** The clinical relevance of noncoding DNA polymorphism of *PON* cluster in stroke.

Clinical Relevance	Polymorphic Site (Functional Alias)	DbSNP Rs Number	References
T allele increased the overall risk of arterial ischemic stroke in young adults (age <45); TT genotype two times increased risk of acute hemispheric stroke in the elderly (age >56)	-107/108 C/T; *PON1* promoter	rs705379	[103,104]
A/G polymorphism associated with risk of total and ischemic stroke	-162 A/G *PON1* promoter	rs705381	[106]
G/A variations were independent predictor of carotid IMT; A allele identified as protective factor	-824/-832 G/A *PON1* promoter	rs854571	[105,106]
CT or TT genotype via miR 616 associated with lower risk of ischemic stroke and lesser extent of carotid atherosclerosis	3′UTR *PON1*	rs3735590	[60]

**Table 4 ijms-22-02137-t004:** The clinical relevance of noncoding DNA polymorphism of *PON* cluster in diabetes.

Clinical Relevance	Polymorphic Site (Functional Alias)	DbSNP Rs Number	References
TT genotype more frequent in diabetic patients with CHD; T allele associated with lower PON1 lactonase activity in neonates of gestational diabetes mellitus(GDM )mothers; TT and CT genotypes more frequent in nondiabetic patients with abnormal glucose control; C/T variations associated with insulin sensitivity; TT genotype associated with proatherogenic lipid profile, macrovascular disease and peripheral neuropathy in diabetics	-107/108 C/T; *PON1* promoter	rs705379	[87,88,112,116,117,118,78]
A allele associated with higher urinary albumin loss in diabetics	-162 A/G *PON1* promoter	rs705381	[119]
GG and GC genotypes associated with diabetes in northwest Indian Punjabi population; GG genotype was protective factor against diabetic retinopathy	-907/909 C/G *PON1* promoter	rs854572	[113]
A allele associated with higher urinary albumin loss in diabetics	-1074 A/G *PON1* promoter		[119]

**Table 5 ijms-22-02137-t005:** The clinical relevance of noncoding DNA polymorphism of *PON* cluster in peripheral artery disease, dementia and renal disease.

Clinical Relevance	Polymorphic Site (Functional Alias)	DbSNP Rs Number	References
Higher frequency of C allele in lupus erythematosus (LE) renal disease; allele C frequency higher in patients with non- progressive IgA nephropathy Allele T more frequent in patients with AD, TT genotype associated with unfavorable lipid profile in patients with AD of late onset; in meta – analysis allele A associated with higher risk of AD, protective role of GG genotype entry 1	-107/108 C/T; *PON1* promoter	rs705379	[124,125,126,127,128,134]
Higher frequency of A allele in LE renal disease; Higher albumin loss in diabetic patients with allele A	-160/-162 A/G *PON1* promoter	rs705381	[124,119]
Higher frequency of G allele in LE renal disease; Higher albumin loss in diabetic patients with allele G	-1074 A/G		[124,119]

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
