# Peer review of "The Relevance of Noncoding DNA Variations of Paraoxonase Gene Cluster in Atherosclerosis-Related Diseases"

_ijms, 2021, doi:10.3390/ijms22042137_

Round 1
Reviewer 1 Report
Descriptive manuscript correctly written and presented. Good introductory background, clearly prepared and presented research theses. An aptly conducted discussion of the topic with minor slips of a rather editorial nature. Formalized conclusions that accurately summarize the outlined research problem. The only thing missing - if possible - is a summary in the form of a statistical description.
Author Response
Dear Sir,
We are very grateful for careful analyzing of our text and valuable comments and suggestions.
- All the paper was carefully revised and the extensive correction of English grammar and style was introduced. All typos, e.g., lacking of spaces between words, references, notation and typing errors were re-checked and corrected.
- We afraid that on the basis of presented data, a summary statistical description will not be possible to prepare.
- According to another Reviewer suggestion we decided to remove the reference 128 from our review, because the article in Polish certainly would limit its availability at the scientific community worldwide.
- In accordance to others Reviewers indications we created four additional figures illustrating 1) the arrangement and regulation of PON1, PON2 and PON3 polymorphic sites affecting the enzymes activity, 2) the physiological function of PONs in the defense against the lipid oxidation 3) the scheme of SNPs involved in atherosclerosis related diseases and 4) metabolic pathways, in which statins impact the PON1 activity.
We hope that introduced modifications and additional illustrations will enrich the review and improve the importance of our manuscript and the language correction will clarify the message and facilitate the better comprehension that will allow its publication.
Yours sincerely
Authors
Reviewer 2 Report
Title: The relevance of non – coding DNA variations of paraoxonase gene cluster in atherosclerosis -related diseases
The broad PON enzymes variability in humans, partly due to differences in genetics, can confer differential susceptibility to diseases because these enzymes have antioxidant and anti-inflammatory properties. Studies associating human paraoxonase family members (PON1, PON2, PON3) with atherosclerosis-related diseases are scattered. This manuscript has the merit of reviewing studies where non-coding sequences of PON genes and atherosclerosis related diseases have been considered. The manuscript is quite well presented and organized, representing a comprehensive and useful overview of PON functional relevance and its physiopathological impact.
Few comments:
- What was the rationale to select articles in English or Polish? The second language selection in search does not have equity. A review article should aim at the scientific community worldwide, which share a common language: English. Besides the relevance of adding Polish in the search is minimal - only one reference listed is in Polish. I recommend authors to reconsider this entry in the search.
- Some paragraphs in the paper would benefit of clarification and/or an English revision. Some examples: p.4, ln 119-123; p.8, ln 260-261; p.15, ln 521-524.
- Check for typos, e.g., lacking of spaces between words, references (e.g., 128), notation (Brophy et al), typing errors “total and IS” in Table 3.
- Authors summarize the relationship of PON cluster polymorphic sites with PON activity and diseases in tables.
However, it would be very useful to illustrate the influence of specific genotypes in the role of PON1, PON2 and PON3 activity, intracellular and extracellular action, other enzyme’s activity/ specific metabolic pathways and clinical relevance, pharmaceuticals resistance, to mention some of the features.
From my point of view illustrations of key roles of PON’s would significantly improve the readability of the paper and its attractiveness, therefore its impact in the academic readership.
Author Response
Dear Sir,
We are very grateful for careful analyzing of our text and valuable comments and suggestions.
- According to Reviewer’s recommendation we decided to remove the reference in Polish from our review article. It was discussed in the initial version because we did not found data concerning abdominal aorta aneurysm in English and the issue had appeared worth of the mention. We couldn’t agree more with the Reviewer that the review should aim at the scientific community worldwide and the citation of the article in Polish certainly would limit it.
- All the paper was carefully revised and the extensive correction of English grammar and style was introduced. Indicated examples were also corrected.
- All typos, e.g., lacking of spaces between words, references, notation and typing errors were re-checked and corrected.
- Apart from the tables summarizing the relationship of PON cluster polymorphic sites with PON activity and diseases we created four additional figures illustrating 1) the arrangement and regulation of PON1, PON2 and PON3 polymorphic sites affecting the enzymes activity, 2) the physiological function of PONs in the defense against the lipid oxidation 3) the scheme of SNPs involved in atherosclerosis related diseases and 4) metabolic pathways, in which statins impact the PON1 activity.
We hope that introduced modifications and additional illustrations will enrich and improve the importance of our manuscript and the language correction will clarify the message and facilitate the better comprehension that will allow its publication.
Yours sincerely
Authors
Reviewer 3 Report
In this review Wysocka et al. give an overview of the polymorphisms and genetic variations associated with the PON gene cluster and how these affect the activity and concentration of PON enzymes. Moreover, they analize the available literature and extract conclusions about the extent to which certain genetic variations of the PON gene cluster might trigger the development of several atherosclerosis-related diseases. Finally, they discuss the implication of PONs polymorphisms on the variable response to treatment observed among atherosclerosis and cardiovascular disease patients.
I personally think this review is interesting since the authors have successfully analyzed the literature on the topic and provide a well-structured summary of the most clinically relevant variations of the PON gene cluster. The thorough effort they have put into this manuscript is evident.
Nevertheless, I must strongly point out that an extensive correction of English grammar and style is required. It is sometimes hard to get the message that the authors try to convey due to an improper use of the language.
For instance:
-Lines 42 and 44: bounded and founded don't exist. Change for bound and found.
-Lines 57 and 58: "Despite of the lack of OP-hydrolyzing activities also later detected PON2 and PON3 by analogy were termed paraoxonases"
Despite of is not used. Either in spite of the lack.. OR despite the lack..
Also, for the sentence to be easily understandable it should be formulated:
In spite of lacking OP-hydrolyzing activity, the later discovered PON2 and PON3 were termed by analogy paraoxonases.
-Lines 75 to 78: "Amongst the most widely investigated promoter poly- morphic sites, including –107/108(C/T), –126(G/C), –160/-162(A/G), –824/832(A/G) and –909 (G/C), only –108(C/T) and –162(A/G) are concerned as functional [27-29] with the greatest effect of –108(C/T) promoter region polymorphism."
Again not a correct formulation, I suggest to rewrite the sentence as follows for the sake of simplicity:
Amongst the most widely investigated promoter poly-morphic sites, including –107/108(C/T), –126(G/C), –160/-162(A/G), –824/832(A/G) and –909 (G/C), only –108(C/T) and –162(A/G) are known to be functional [27-29], being the greatest effect caused by the -108(C/T) polymorphism on the promoter region.
-Lines 91 to 97: "Taking into account, that polymorphic variations of PON cluster gene due to direct affecting PONs’ activity and concentration or influencing the regulatory elements binding may contribute to atherosclerosis–related diseases, involving coronary artery disease, stroke, peripheral arteries disease, diabetes mellitus, dementia and renal disease, this issue is worth of the detailed discussion, especially as it was suggested, that evaluation of both, the PON genotype and enzyme activity, may serve as possible predictor of the clinical forms of atherosclerosis [40].
This sentence is hard to understand, making it so long prevents the reader from focusing on the important information that is transmitted. I suggest to rephrase it as follows:
Taking into consideration that polymorphic variations of the PON cluster gene directly affect PON's activity, concentration and the binding of regulatory elements, it is reasonable to believe that they may contribute to atherosclerosis–related diseases such as coronary artery disease, stroke, peripheral arteries disease, diabetes mellitus, dementia and renal disease. Indeed, this issue is worth discussing since both, the PON genotype and enzyme activity, may serve as possible predictors of the clinical forms of atherosclerosis [40].
Unfortunately, the incorrect use of English language is consistent throughout the review and poses a barrier for understanding its relevant content. Therefore, my main suggestion is to revise the full text and implement grammar and sentence formulation mistakes.
Apart from that, I would encourage the authors to create a figure depicting the function of paraoxonases in lipid profile regulation. It would certainly enrich the review and facilitate a better comprehension of the importance of these enzymes.
Author Response
Dear Sir,
We are very grateful for careful analyzing of our text and valuable comments and suggestions.
- According to the Reviewer’s main suggestion the full text were carefully revised and the extensive correction of English grammar and style was implemented. The sentences found as difficult for understanding were rephrased in accordance to Reviewer ‘s indications.
- We created a figure depicting the physiological function of paraoxonases in the defense against lipid oxidation and additionally three other figures illustrating the arrangement and regulation of PON1, PON2 and PON3 polymorphic sites affecting the enzymes activity, the scheme of SNPs involved in atherosclerosis related diseases and metabolic pathways, in which statins impact the PON1 activity.
- According to another Reviewer suggestion we decided to remove the reference 128 from our review, because the article in Polish certainly would limit its availability at the scientific community worldwide.
We hope that introduced modifications and additional illustrations will improve the importance of our manuscript and the language correction will clarify the message and facilitate the better comprehension that will allow its publication.
Yours sincerely
Authors
Round 2
Reviewer 3 Report
Thank you for taking into consideration my comments. I believe that the changes have greatly improved the quality of this review and therefore will gladly reccommend its publication. Congratulations for this work!